# Negentropy anomaly analysis of the borehole strain associated with the Ms 8.0 Wenchuan earthquake

Kaiguang Zhu[1,2], Zining Yu[1,2], Chengquan Chi[1,2], Mengxuan Fan[1,2], and Kaiyan Li[1,2]

[1]College of Instrumentation and Electrical Engineering, Jilin University, China
[2]Key Laboratory of Geo-Exploration Instrumentation, Ministry of Education, Jilin University, China

**Correspondence:** Kaiguang Zhu (zhukaiguang@jlu.edu.cn)

**Abstract.** A large earthquake of 8.0 magnitude occurred on 12 May 2008, 14:28 UTC, with the epicenter in Wenchuan. To investigate the pre-earthquake anomalous strain changes, negentropy is introduced to borehole strain data for 3 locations, approximated by skewness and kurtosis revealing the non-Gaussianity of recorded fluctuations. We separate the negentropy anomalies from the background by Otsu's method and accumulate the anomaly frequency in different scales. The results show the long-scale cumulative frequency of negentropy anomalies follows a sigmoid behaviour, while the inflection point of the fitting curve is close to the occurrence of the earthquake. For the short-scale analysis before the earthquake, there are two cumulative acceleration phases. To further verify the correlation with the earthquake, we compare our findings for different time periods and stations, and rule out the possible influence of meteorological factors. We consider the negentropy analysis exhibits potential for studying pre-earthquake anomalies.

## 1 Introduction

Changes in crustal deformation fields over time have been recorded at least for some large earthquakes (Thatcher and Matsuda, 1981), such as the 2011 Tohoku earthquake (Hirose, 2011) and the Ruisui earthquake in Taiwan in 2013 (Canitano et al., 2015). Borehole strainmeters which detect the crustal changes provide an opportunity to investigate preparation process prior to earthquakes. Many strain observations were of research significance (Linde et al., 1996), although there were some unsuccessful detections, such as 1987 Superstition Hills earthquake (Agnew and Wyatt, 1989), 1989 Loma Prieta earthquake (Johnson et al., 1990) and 2009 L'Aquila earthquake (Amoruso and Crescentini, 2010). Various methods are used in identifying borehole strain anomalies based on large amount of monitoring data. Experienced scholars extract borehole strain anomalies by discriminating patterns of waveform behaviors compared to those during the normal stage (Johnston et al., 2006; Chi et al., 2014). In the time domain, Qiu et al. (2011) identified abnormal strain changes by overrun rate and wavelet decomposition for the Wenchuan earthquake. While in the frequency domain, Qi and Jing (2011) thought the signal with a period of 10 to 60 minutes might be anomalies through S-transform compared with the background signal. In addition, statistical methods are proved effective in distinguishing borehole strain anomalies with regard to large earthquakes, such as principal component analysis (Zhu et al., 2018) and correlation coefficients along with the consistency relation (Kong et al., 2018).

The probability distribution function (PDF) of observation data is also an informative way of extracting potential anomalies contained in earthquake generation processes. Manshour et al. (2009) extracted variance anomalies of the probability density of the Earth's vertical velocity increments, and successfully found a pronounced transition from Gaussian to non-Gaussian prior to 12 moderate and large earthquakes. Before the Wenchuan earthquake, the high-frequency fluid observational data deviated from Gaussian distributions at 16 water level and 14 water temperature stations (Sun et al., 2016).

Rather than the whole PDF, often its moments are utilized, moments may be estimated quite reliably from relatively small amounts of data (Sattin et al., 2009). In 2016, Chen applied skewness and kurtosis (the third- and fourth-order moments) of the geoelectric data to pick up non-Gaussian distribution anomalies to predict impending large earthquakes in Taiwan. On the other hand, for turbulent or disordered systems, the non-Gaussian distribution of time series in skewness-kurtosis domain attracts attention. Observation data series from various fields of geophysics indicate that a parabolic relation between skewness and kurtosis holds in fields such as seismology (Cristelli, 2012), oceanography (Sura and Sardeshmukh, 2012) and atmospheric science (Maurizi, 2006).

Thereby, it is possible that precursor anomalies lead to an increase of disordered components in observation data. Eftaxias et al. (2015) proved that the pre-catastrophic stage could break the persistency and high organization of the electromagetic field through studying fractional-Brownian-motion-type model using laboratory and field experimental electromagetic data. In view of Lévy flight and Gaussian processes, Lévy flight mechanism prevents the organization of the critical state to be completed before earthquakes, since the long scales are cut-off due to the Gaussian mechanism (Potirakis et al., 2019).

Entropy can serve as a measure of the unknown external energy flow into the seismic system (Akopian, 2015). Karamanos et al. (2006, 2005) quantified and visualized temporal changes of the complexity by approximate entropy, they claimed significant complexity decrease and accession at the tail of the preseismic electromagnetic emission could be diagnostic tools for the impending earthquake. Approximate entropy is also been studied in catastrophic events (Nikolopoulos et al., 2004). Ohsawa (2018) detected earthquake activation precursors by studying the regional seismic information entropy on earthquake catalog.

Negentropy definition is based on the entropy and it is also widely used to detect non-Gaussian features. **?** proposed an arrival-time picking method based on negentropy for microseismic data. In this study, the negentropy is applied to borehole strain at Guza station associated with the Wenchuan earthquake, approximated by skewness and kurtosis revealing the non-Gaussianity of borehole fluctuations. Subsequently we study the extracted negentropy anomalies in different scales to investigate correlations with crustal deformation. Furthermore, we did a comparison discussion for different time periods and stations.

## 2   Observation

YRY-4 borehole strainmeters, which are designed to record continuous deformation occurring over periods of minutes to years, have been deployed at depths of more than 40 metres at more than 40 terrain-sensitive locations within China. These strainmeters are capable of resolving strain changes of less than one-billionth. The data sampling rate is once per minute.

The study period is from January 1, 2007, to June 30, 2009. The study area is shown in Figure. 1. We find the Guza station stands on the southwestern end of the Longmenshan fault zone. Besides, the epicentre is about 150km away from the station, which is within the monitoring capability of the borehole strainmeters (Su, 1991).

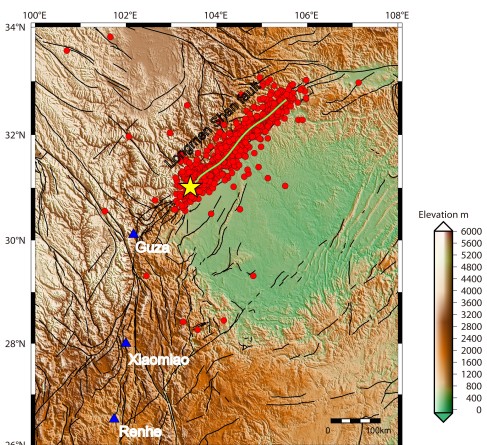

**Figure 1.** Location map showing the epicentre and three stations. The epicentre was located at $31.01°$N, $103.42°$E shown by yellow star. The blue rectangles show the strainmeter stations. The red circles are aftershock distribution from the main shock on May 12, to October 30, 2008. The green curve is the schematic curve of the main rupture zone and black curves are faults.

Because the four gauges of the YRY-4 borehole strainmeter are arranged at $45°$ intervals, this design has improved its self-consistency. This arrangement produces four observation values: $S_i$, $(i = 1, 2, 3, 4)$ (Qiu et al., 2013). The self-consistency as shown in equation (1), which can be used to test the reliability of the data among the four gauges.

$$S_1 + S_3 = S_2 + S_4 \tag{1}$$

In practical application, the higher the correlation between both sides of the equation (1), the more reliable the data. In that case, we can use areal strain $S_a$ for describing the subsurface strain state instead of four component observations. $S_a$ is expressed as

$$S_a = (S_1 + S_2 + S_3 + S_4)/2. \tag{2}$$

The borehole strain of Guza station is highly consistent (Qiu et al., 2009) as shown in Figure. 2. Then we processed the daily areal strain through two steps.

Step 1: Differential calculation.

We set the areal strain data as $X(n)$ and differential areal strain data as $Y(n)$, we know $Y(n) = X(n) - X(n-1)$, where $n$ is the sample point. The process can be equivalent to a filtering system, $H_1(e^{j\omega})$ is frequency responses of the Step 1,

$$\mid H_1(e^{j\omega}) \mid = \sqrt{2(1 - cos\omega)}. \tag{3}$$

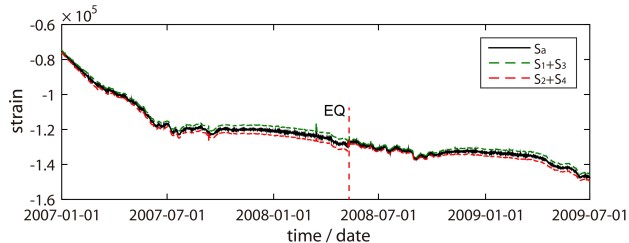

**Figure 2.** Self-consistency of the borehole strain at Guza from January 1, 2007, to June 30, 2009.

It can be seen that when $\omega$ is very small or 0, the frequency response is 0, indicating that the Step 1 removes the low frequency information of the signal, including borehole trend and low frequency effects of the air pressure and temperature on the signal.

Step 2: Harmonic analysis.

We remove the periodic term that still exists through daily harmonic analysis. We set the fitting function as Fourier series. The reserved signal $Z(n)$ can be simplified as $Z(n) = Y(n) - \sum_{k=1}^{n} A(k)sin(k\omega_0 n + \varphi_i)$, $k\omega_0$ and $\varphi_i$ are frequencies and phases of the periods of the daily data, $A(k)$ is corresponding amplitudes.

$H_2(e^{j\omega})$ are frequency responses of the Step 2, by minimizing $Z(n)$ through least squares method in time domain, then ideally for the frequency response is

$$| H_2(e^{j\omega}) | = \begin{cases} 0 & \omega = k\omega_0, \\ 1 & \omega = others. \end{cases} \tag{4}$$

The Step 2 removes the periodic terms in the signal. We think the period terms mainly includes the periods related to the solid tide, also includes the periodic effects of air pressure. The residual high-frequency signals are shown in Figure. 3. In particular, small changes in the curve are amplified by the processing.

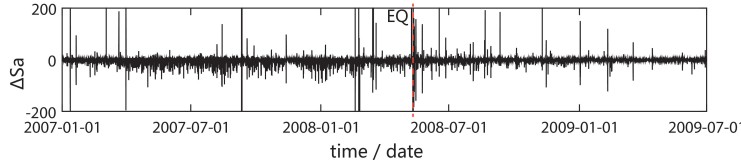

**Figure 3.** High-frequency areal strain at Guza from January 1, 2007 to June 30, 2009.

## 3 Methodology

### 3.1 Negentropy and non-Gaussianity

The entropy-based negentropy is a statistically justified measure of non-Gaussianity (Hyvarinen and Oja, 2000). The entropy of a random variable $X = \{x_1, x_2, ..., x_i, ...\}$ is defined as

$$H(X) = -\sum_i P(X = x_i) \log P(X = x_i), \tag{5}$$

where $P$ is the probability density function. Entropy measures the randomness of a random variable. The Gaussian random variable has the largest entropy of all other random variables with equal variance (Cover and Thomas, 1991). The definition of negentropy is given by

$$J(X) = H(X_{guass}) - H(X), \tag{6}$$

in which $X_{gauss}$ is a Gaussian random variable with the same mean and covariance matrix as $X$. The entropy of a Gaussian random variable can be estimated by

$$X_{guass} = \frac{1}{2} \log |\det \Sigma| + \frac{n}{2}(1 + \log 2\pi), \tag{7}$$

where $n$ is the dimension of the variable, and $\Sigma$ is its covariance matrix.

However, the theoretical calculation of negentropy also depends on the prior probability density of random variables and other information which are difficult to determine accurately. In practical applications, higher order statistics (HOS) and density polynomial expansion are usually used to approximate one-dimensional negentropy (Jones and Sibson, 1987). The approximation results are as follows:

$$J(X) \approx \frac{1}{12} skewness^2(X) + \frac{1}{48} kurtosis^2(X). \tag{8}$$

This definition suggests that any deviation from a Gaussian distribution will increase the negentropy $J(x)$. The skewness and kurtosis are the third- and fourth-order statistics, respectively, which are defined as

$$skewness(X) = \frac{\mu_3}{\sigma^3} = \frac{E[(X - \mu)^3]}{E[(X - \mu)^2]^{3/2}} \tag{9}$$

and

$$kurtosis(X) = \frac{\mu_4}{\sigma^4} = \frac{E[(X - \mu)^4]}{E[(X - \mu)^2]^2} - 3, \tag{10}$$

where $\mu$ is the mean of $X$ and $\sigma$ is the standard deviation of $X$. Skewness is a measure of asymmetry in a PDF. A symmetric distribution has zero skewness. Kurtosis is a measure of the heaviness of the tails. Distributions that are more outlier-prone than the normal distribution have kurtosis values greater than zero.

Moreover, the relation between the skewness and kurtosis is universal, they approximately align along a quadratic curve (Sattin et al., 2009):

$$kurtosis(X) = A \cdot skewness^2(X) + B. \tag{11}$$

Here we calculate the normalized skewness and kurtosis in the study period, so equation (9) can be derived into

$$kurtosis(X) = A \cdot (skewness^2(X) - 1), \tag{12}$$

indicating the test day is super-Gaussian when the skewness is outside the range (-1,1).

This relation is trivial in a Gaussian fluctuating system; it reduces to a fixed mass around zero (skewness=0 and kurtosis=0). In a turbulent environment where fluctuating quantities obey non-Gaussian statistics, the moments obey the above relation.

## 3.2 Otsu's thresholding method

To solve the negentropy anomaly detection problem, we designed a simple thresholding hypothesis test using the Otsu method (Otsu, 1979) that provides an optimal separation between background and seismic-related activities. For any given value k, we can separate the previously calculated $J(x)$, as shown in equation (6), into the following two classes:

$$
\begin{aligned}
C_0(k) &= \{J(x) \le k\}, \\
C_1(k) &= \{J(x) > k\}.
\end{aligned}
\tag{13}
$$

Using these classes, the weighted average value $\mu_T(x)$ of $J(x)$ can be expressed as follows:

$$
\begin{aligned}
\mu_T(x) &= \lambda_0(k)\mu_0(x;k) + \lambda_1(k)\mu_1(x;k), \\
\lambda_0(k) &+ \lambda_1(k) = 1.
\end{aligned}
\tag{14}
$$

where $\mu_0(x;k)$, $\mu_1(x;k)$ are the mean values of the class $C_i(k)$, $i$=0, 1, and $\lambda_i(k)$ is the percentage of points belonging into each class. Following the thresholding scheme of Otsu (1979), we define the following cost function:

$$
\begin{aligned}
\sigma_B{}^2 &= \lambda_0(k)(\mu_0(x;k) - \mu_T(x;k))^2 + \lambda_1(k)(\mu_1(x;k) - \mu_T(x;k))^2, \\
&= \lambda_0(k)\lambda_1(k)(\mu_1(x;k) - \mu_0(x;k))^2.
\end{aligned}
\tag{15}
$$

where $\sigma_B{}^2$ is the within-class variance of negentropy. Then, by finding the $k^*$ value searching for $k$ when $\sigma_B{}^2$ becomes the maximum

$$k^* = \arg\max_k \sigma_B{}^2(k), \tag{16}$$

the optimal value $k^*$ here separates the background set and anomaly set.

In this test, our initial assumption is that the sliding window is composed of a Gaussian signal of non-seismic-related activities. When our test negentropy exceeds the critical value k*, this initial hypothesis is not valid, and the alternative is true, indicating the presence of a negentropy anomaly within the window.

## 4    Results

According to the empirical hypothesis that geophysical signals deviate from the Gaussian distribution when they record abnormal activities, and based on the results of previous studies, we perform the following investigation.

### 4.1    Extracting negentropy anomalies

As the negentropy is calculated using a 2-hour long sliding window, we assume that it reaches the maximum values when the time window contains anomalies from seismic-related activities. The negentropy during the study period is shown in Figure. 4.

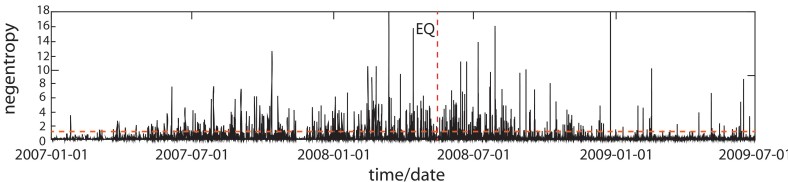

**Figure 4.** Negentropy at Guza from January 1, 2007, to June 30, 2009. The red dotted, horizontal line is the optimal threshold $k^*$ calculated by Otsu method.

The within-class variance $\sigma_B{}^2$ and negentropy value distribution are compared in Figure. 5. According to equations (11) to (14), when $k^* = 1.1130$, $\sigma_B{}^2$ reaches its maximum. Therefore, the negentropy were separated by $k^*$ into the quasi-Gaussian background and non-Gaussian anomalies from 2007 to 2009. Otsu threshold $k^*$ here is consistent with the accuracy of the negentropy and the strain data. The YRY-4 borehole strainmeter has a measurement accuracy of $10^{-9}$, however, we usually cutoff four digits after the decimal point in practical calculations.

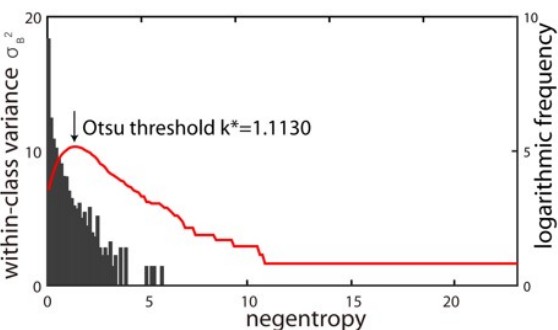

**Figure 5.** Within-class variance $\sigma_B{}^2$ of the negentropy (red line) and negentropy histogram.

In the skewness-kurtosis domain, the statistical relationship of the borehole areal strain is consistent with parabolic behaviour as described in equation (10) (Figure. 6(a)), verifying that the turbulent system of borehole strain is significantly non-Gaussian. Besides, the extracted negentropy anomalies are clustered strongly on the left side of the parabola, which exhibit similar

characteristics different from the normal Gaussian distribution. Here, there are four points on the right side; one occurred in early 2007, and the others occurred after the earthquake. Therefore, we will not discuss them in the following.

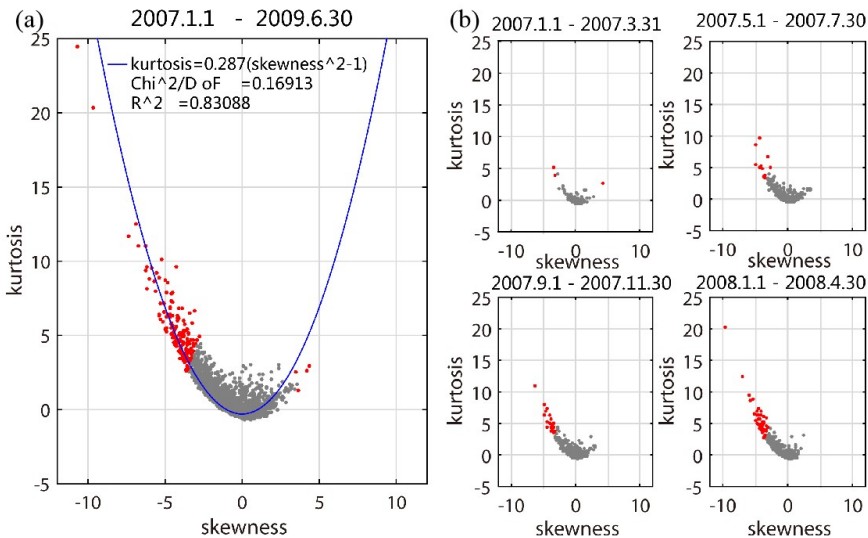

**Figure 6.** Negentropy distributions in the skewness-kurtosis domain in (a) January 1 ,2007, to June 30 2009, and (b) four shorter periods before the earthquake. Red denotes that the negentropy is greater than $k^*$, and grey indicates that the negentropy less than $k^*$. The blue curve is the quadratic fit with a $95\%$ confidence.

In addition, as shown in Figure. 6(b), at times far from the earthquake, the negentropy distribution is basically Gaussian in the skewness-kurtosis domain. However, at times closer to the earthquake, the relatively stable state was broken due to the non-Gaussian mechanism, with more negentropy anomalies appearing on the left side of the parabola. While in 2008, almost all of the negentropy present left-skewed.

The phenomena prompt us to study its possible correspondence with the seismogenic process.

## 4.2 Negentropy anomaly frequency accumulation

The transition of negentropy anomalies in the skewness-kurtosis domain is quantified as the change of the anomaly frequency per unit time through a logarithmic-linear model. Logarithmic-linear models are often used of interest to estimate the expected frequency of the response variable at the original scale for a new set of covariate values, such as the Gutenberg-Richer law, in which a linear relationship exists between the logarithm of the cumulative number of seismic events of magnitude $M$ or greater versus the magnitude $M$ (Gutenberg and Richter, 1954).

The logarithmic-linear regression model is proposed as

$$\log N = \beta_1 \times k_J + \beta_0 + \varepsilon, \tag{17}$$

where $k_J$ takes different threshold values according to the $J(x)$ values, $N$ is the number of occurrences in which $J$ is greater than or equal to threshold $k_J$, $\beta_1$ and $\beta_0$ are the regression coefficients, where a lower slope $\beta_1$ indicates that there are more higher $J$ values, implying there are more anomalies at that moment, and $\varepsilon$ is the random error that represents the model uncertainty.

We use the logarithmic-linear model to solve the relationship between the negentropy anomaly frequency and different thresholds each day using the ordinary least squares method (OLS) method. Afterwards, an optimal threshold $k^*$, calculated by the Otsu method, is chosen for all models, where

$$N_J(t) = \exp(\beta_1(t) \times k^* + \beta_0(t) + \varepsilon(t)) \tag{18}$$

and the $N_J(t)$ under the threshold $k^*$ is shown in Figure. 7. The model theoretically solves the problem of selecting the length of the time window. In addition, the estimated $N_J(t)$ is considered as the expected frequency of anomalies.

The goodness of fit for each logarithmic-linear model was evaluated using analysis of

$$R^2 = 1 - \sum_{i=1}^{n}(N_i - \hat{N}_i)^2 / \sum_{i=1}^{n}(N_i - \overline{N_i})^2 \tag{19}$$

and the root-mean-squared error(RMSE).

$$RMSE = \sqrt{\sum_{i=1}^{n}(N_i - \hat{N}_i)^2/n}. \tag{20}$$

The $R^2$ and RMSE values in the study period (912 days) show that the logarithmic-linear relationship can explain the relationship between the negentropy anomaly frequency and different thresholds. The mean of $R^2$ is 0.9695, which is close to 1, and the variance of $R^2$ is 0.0435. The mean and variance of the RMSE are also small (0.1098 and 0.1301, respectively).

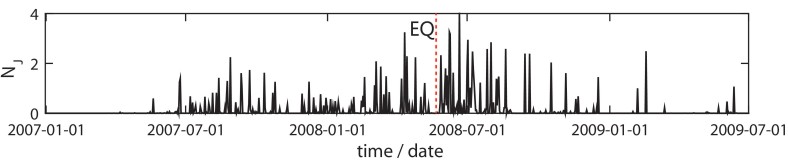

**Figure 7.** Estimated expected frequency $N_J$ under the optimal threshold $k^*$

We calculate the negentropy cumulative frequency of the study period as shown in Figure. 8. There is not only a long-scale analysis of the whole period, but also a short-scale analysis of the pre-earthquake process. In general, accumulated value of a typical random process usually has a linear increase. In particular, in case of critical phenomena, we would expect more frequent anomalies when they approach the critical point, and less frequent anomalies after (Santis et al., 2017).

For the entire earthquake process, a two-month long sliding window is selected for accumulation. In Figure. 8(a), after July 2007, the negentropy anomalies gradually accumulated. Qiu (2009) and Chi (2014) also observed anomalies of this period at

Guza station, they speculated that abnormal strain may reflect small-scale rock formation rupture before the earthquake. In particular, we find more frequent negentropy anomalies in 2008 as the earthquake approaches, and less frequent anomalies after, so a sigmoid function is used to fit the acceleration, before the earthquake and the deceleration after. Sigmoid function is expressed as

$$y = A2 + \frac{(A1 - A2)}{(1 + e^{\frac{x - x_0}{dx}})},$$ (21)

where $A1$, $A2$, $x0$ and $dx$ is the inflection point. The sigmoid function is a power-law temporal behavior with an upper concavity and a subsequent power-law behavior after the inflection point, with an opposite concavity. The inflection point in this function is a reasonable estimation of the time of the significant change in the critical dynamical system (Santis et al. , 2017). Also, the value of x0 (8.3337) obtained in the fitting result is almost coincide with the Wenchuan earthquake day. The actual time of the earthquake is 8.3871 after conversion.

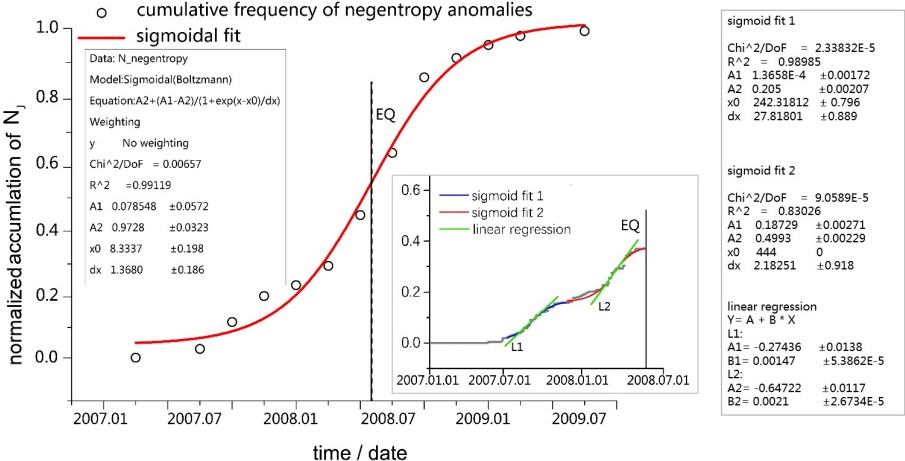

**Figure 8.** (a) Results of the long-scale negentropy anomaly frequency for the Wenchuan earthquake at Guza station from January 1, 2007, to June 30, 2009. Each circle represents an anomaly negentropy for 2 months. The cumulative frequency of negentropy anomaly is represented. The earthquake day is represented as a vertical black dashed line. The red line is a sigmoid fit that underlines an inflection point (vertical solid line) is close to the occurrence of the earthquake. (b) Results of the short-scale negentropy anomaly frequency prior to the earthquake, every grey point is an anomaly for one day, the blue and red lines are two segment sigmoid fit results. Two green lines represent the liner regression for the two phases, the first phase slope is 0.00147, the second one is 0.0021.

When we narrowed the accumulated window to one day, we observed two negentropy anomalies before the earthquake as shown in Figure. 8(b). The first anomaly frequency increase occurred from August to October 2007. In March 2008, there was a second phase of anomaly increase, and the cumulative frequency then slowly increased to a plateau period near the time of the earthquake. This probably due to the stress is in deadlocked phase. Since before the Wenchuan earthquake, the elastic deformation of the crust reaches its limit and the deformation is resisted in the hypocentral region, which is measured by GPS data (Jiang et al., 2009).

These two phases prior to the earthquake are also approximated with sigmoid functions. In order to further compare the anomalies of the two phases, we use linear regression to fit the central part of the two sigmoid curves. We find that the second acceleration is greater than the first acceleration.

     Fault zones contain relatively weak and relatively strong parts. The former is the area where strain release begins, while the latter is the stress locking part and the beginning of rapid instability (Noda et al., 2013). Ma and Guo (2014) proposed there is
a sub-instability stage of fault deformation before earthquakes, which is manifested by two instability activities. The former is related to the release of weak parts, and the latter is related to the rapid release of strong parts during strong earthquakes. She thought acceleration of the strain release in fault zone is a sign of entering the inevitable earthquake stage. Thus, we speculate that the two accelerations of the cumulative negentropy anomaly in Figure. 8(b) may be related to the strong earthquake.

## 5    Comparison discussion

**5.1    Comparison of random time periods.**

We randomly selected the strain data for 200 days before and after March 20, 2011 and March 24, 2014 at Guza station. The selected data for the two periods are required to be in the absence of strong earthquakes and with higher quality. We performed negentropy analysis on these two observations and compared them with the results of negentropy analysis associate with the Wenchuan earthquake as shown in Figure. 9.

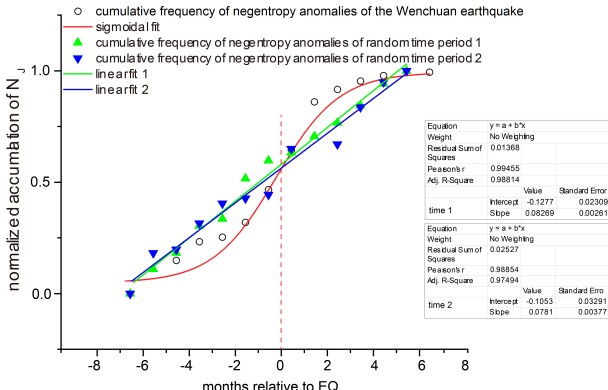

**Figure 9.** The comparative analysis of cumulative frequency of negentropy anomalies between earthquake period and random time periods. The zero point of green dots is March 20, 2011, the zero point of blue dots is March 24, 2014

As we can see in Figure. 9, The cumulative frequency of negentropy anomalies of random periods have linear increase. However, in the Wenchuan earthquake periods, as the earthquake approaches, the cumulative frequency of negentropy anomalies increases rapidly and recovered to a slow growth after the earthquake.

## 5.2 Comparison of different stations

We selected Xiaomiao station and Renhe station to find out if their observations received strain changes. Their locations are shown in Figure. 1. Compared with the Guza station, we did the negentropy analysis of these two stations as shown in Figure. 10.

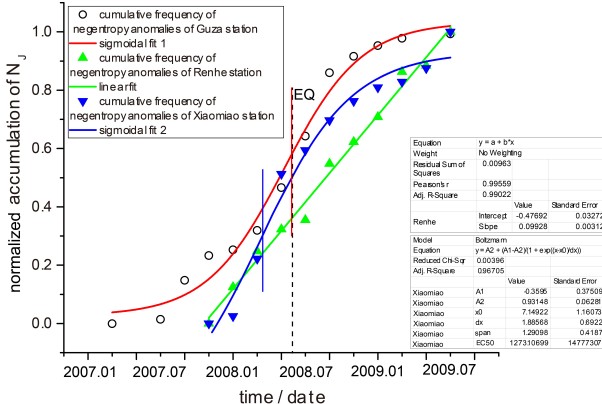

**Figure 10.** Cumulative frequency of negentropy anomalies of Xiaomiao station and Renhe station from September 16, 2007 to June 30, 2009. The negentropy analysis of Guza station is from January 1, 2007 to June 30, 2009, because of the different installation time of the instruments. The red vertical line is the inflection point of the fitting curve of Guza station. The blue vertical line is the inflection point of the fitting curve of Xiaomiao station. The black dotted line is the earthquake day.

As we can see in the Figure. 10, the cumulative frequency of negentropy anomalies of Xiaomiao station are also well fitted by the sigmoid function. The accumulation curve is growing rapidly before the earthquake and concave downward after which is similar to the Guza station, although the inflection point of Xiaomiao station is about two months preceding the earthquake moment. However, since the curve is approximately linear before and after the inflection point, we consider that the inflection point value is reasonable in the range from January to June, 2008. Cumulative anomalies of the Renhe station are basically linear, indicating that the Renhe station may not detect pre-earthquake anomalies.

Renhe station is far from the end of the Wenchuan earthquake fault, so it is reasonable that no abnormal changes are observed. However, Xiaomiao station is located between the Guza station and Renhe station, and the fitting result shows that there is a similar trend to the Guza station, with a weaker curvature. So, for the nearest station to the epicentre, Guza station may be able to record more pre-earthquake anomalies.

Furthermore, Qiu et al. (2012) found that the anomalies at Ningshan station were similar to the anomalies at Guza station. Such two stations have observed similar Wenchuan earthquake precursor anomalies, which may not be accidental. Since the Ningshan station is actually located at the northeastern end of the Longmenshan fault zone. This location is a correspondence with the southwestern end of the fault where the Guza station is.

### 5.3 Exclusion of meteorological factors.

The strain signals are sensitive to a few meteorological factors, therefore, we display the pressure variations, temperature variations recorded at Guza station and the daily rainfall measured by Tropical Rainfall Measuring Mission (TRMM) satellite which are downloaded through the NASA GIOVANNI-4 for the same period and the same area (http://giovanni.gsfc.nasa.gov/giovanni/) in Figure. 11. There are clearly annual variations in the strain, air pressure, temperature and rainfall data. The air pressure and temperature have been steadily fluctuating within a certain range, and the rainfall is also shown to be more in summer and less in winter.

While we calculated the differential data of the strain for negentropy analysis. So, we make differential calculations for all three influencing factors as shown in Figure. 12.

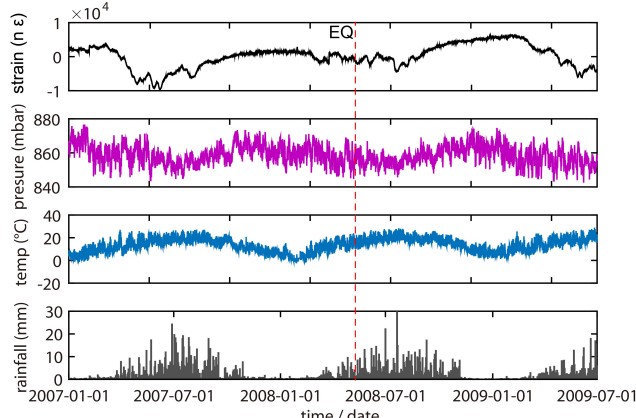

**Figure 11.** Borehole stain, air pressure, temperature and rainfall variations during study period at Guza station.

We observed that the air pressure, temperature and rainfall didn't change abnormally during the period when the extracted anomalies increase, whether we do differential calculation. Therefore, we consider that the abnormal variations on the processed strain signals are not caused by these factors.

### 6  Conclusions

In our work, the extracted negentropy anomalies of borehole strain associated with the Wenchuan earthquake are analyzed. The cumulative frequency of negentropy anomalies are studied in both long- and short-scale. In Comparison discussion, we compare the cumulative anomalies of different time periods and different stations with those at Guza station during the study period, and preliminarily exclude meteorological factors. We suspect the negentropy anomalies at Guza station may have recorded abnormal changes related to the Wenchuan earthquake.

Since the tectonic dynamics of earthquakes during seismogenic and seismic processes are very complex, the mechanism of such abnormal changes is undoubtedly needed to be discussed. In particular, borehole strain signals are sensitive to external

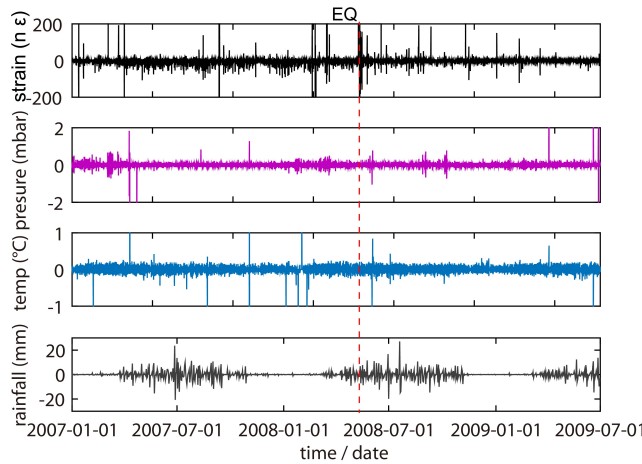

**Figure 12.** Differential borehole stain, air pressure, temperature and rainfall variations during study period at Guza station.

influence. Besides, because of the different characteristics and accuracy of different types of observations, a joint analysis has not been carried out yet. Further researches are needed to decipher a potential precursory phase. However, we may be able to ensure that the negentropy analysis has great potential in the study of earthquake precursors.

*Data availability.* The borehole strain data is a confidential information and therefore cannot be made publicly accessible. The air pressure
and temperature data can be downloaded from National Earthquake Precursor Networks Center only through approval (qzweb.seis.ac.cn).
The rainfall data are downloaded through the NASA GIOVANNI-4 (http://giovanni.gsfc.nasa.gov/giovanni/).

*Author contributions.* The authors contributed in accordance with their competence in the research subject. The first author Kaiguang Zhu was responsible for the key technical guidance and ideas. Zining Yu was responsible for method improvement, data analysis and manuscript preparation. Chengquan Chi helped to ensure the graph quality of the manuscript. Mengxuan Fan and Kaiyan Li contributed through active
participation in the manuscript preparation.

*Competing interests.* The authors declare that they have no conflict of interest.

*Acknowledgements.* The authors would like to thank the China Earthquake Network Center providing the borehole strain data and NASA Giovanni team for rainfall data. Moreover, the authors are grateful to Professor Qiu Z. H. for his guidance and helpful suggestions. This research was supported by the Institute of Crustal Dynamics, China Earthquake Administration (Grant No.3R216N620537).

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
