# Peer review of "Negentropy anomaly analysis of the borehole strain associated with the Ms 8.0 Wenchuan earthquake"

_Nonlinear Processes in Geophysics, 2019_

## Short Comment (SC1) · 24 Jun 2019

In Introduction (L. 19-21), the authors stated that Canitano et al. (2015) observed changes in the crustal deformation prior to the October 2013 Ruisui earthquake, Taiwan. There is a possible confusion here, Canitano et al. (2015) analyzed the strain signals 10 s before the rupture but found no evidence for preseismic strain changes (at least at the instrumental noise level $\sim$ 0.01-0.1 s nstrain).

Alexandre Canitano

2019-22, 2019.

---

## Author Comment (AC1) · 26 Jun 2019

We are very grateful for your short comment.

This is an important issue. Thanks for your suggestion, this Introduction (L. 19-21) is indeed misleading because of the inappropriate word "preceded". In fact, we intended to express the crustal changes have been recorded when some large earthquakes occurred. Therefore, it could provide a great opportunity to study earthquake preparation processes.

In this regard, we modify the Introduction (L. 19-20) as follows:

[Figure]

"Changes in crustal deformation fields over time have been record at least some large earthquakes (Thatcher, W. et al., 1981),..."

[Figure]

---

## Referee Comment (RC1) · Anonymous Referee #1 · 23 Jul 2019

Referee Report concerning NPG submission:

npg-2019-22

Title: Negentropy anomaly analysis of the borehole strain associated with the Ms 8.0 Wenchuan earthquake Author(s): Kaiguang Zhu et al.

Recommendation: Consideration Under Major Revision.

General Comments.

This is an interesting overall manuscript addressing the problem of the earthquake precursors, through the use of the so-called borehole strain method and standard sta-

tistical tools. The findings and principal outcomes of the manuscript are meaningful and important. In any case, I am afraid that the authors are ignorant of some important results in the literature (please consult the "Bibliographical Comments" below). In my opinion, the idea and realization is certainly interesting, and I recommend the manuscript for publication, if the authors address properly, or at least in a satisfactory manner the scientific and technical issues raised below.

Specific Comments: Important Remarks about the Scientific Content.

- The method of detection of anomalies of the borehole strain, is not well-known to non-specialists, and – I would say – to the specialists neither. In my opinion, at least one additional explanatory paragraph entirely devoted to this subject is needed, in the "Introduction" Section.

- In my opinion, in Fig. 6, "kurtosis=0.28699skewnessˆ2-0.28696" should boil down to "kurtosis=0.287(skewnessˆ2-1)", I mean that in equation (9), A=B which is a Remarkable result, if it holds true !!! At least one additional explanatory paragraph entirely devoted to this result is needed, in the "Discussion and Conclusions" Section !

- In line 153, is stated that "k*=1.1130". What is the meaning of keeping so many significant digits ? Why not " k*=1.1 " or "k*=1.11" ? Please explain ! At least one additional explanatory paragraph is needed !

- In line 157, Fig.5, explain the Units !

Bibliographical Remarks.

I think the authors could find interesting – and include in their list of References – those quite old works, one of them published in NPG:

- "Extracting preseismic electromagnetic signatures in terms of symbolic dynamics." K.Karamanos, A. Peratzakis, P. Kapiris, S. Nikolopoulos, J. Kopanas and K. Eftaxias Nonlinear Processes in Geophysics 12, 835-848 (2005) - "Study of pre-seismic electromagnetic signals in terms of complexity." K. Karamanos, D. Dakopoulos, K. Aloupis, A.

Peratzakis, L. Athanasopoulou, S. Nikolopoulos, P. Kapiris and K. Eftaxias Phys. Rev. E 74, 016104 – 016125 (2006)

- "Evidence of fractional-Brownian-motion-type asperity model for earthquake generation in candidate pre-seismic electromagnetic emissions." K. Eftaxias, Y. Contoyiannis, G. Balasis, K. Karamanos, J. Kopanas, G. Antonopoulos, G. Koulouras and . Nomicos Nat. Haz. Earth Syst. Sci. 8, 657-669 (2008)

A more recent Reference could be for instance:

- "Levy and Gauss statistics in the preparation of an earthquake." S.M. Potirakis, Y. Contoyiannis and K. Eftaxias Physica A, Vol. 528, 15 August 2019, 121360 (In Press)

Technical Comments and Error Corrections.

In my opinion however, the authors did not spend enough time for the preparation of their manuscript, so that plenty of Minor corrections are needed !!!

For instance:

- In line 17, "earthqake" → "earthquake" !

- In lines 26 and 27, the citation has no uniform style !

- In lines 295 and 296, of the References list, there are quotation marks in the title of the Reference. This is the only place in the whole list where this happens !

- In line 153, there are superscripts in the middle of the sentence, for no reason !

- In line 157, the end dot (final punctuation mark) is missing !

- In line 159, it is mentioned "Fig 6(a)" instead of the correct "Fig. 6(a)" (the dot is missing) !

- In line 297, "Gutenber" → "Gutenberg" !

- In line 325, the style is not uniform ! Dots are missing !

- In line 341, the style is not uniform ! Dots are missing !

- In line 349, the style is not uniform ! Dots are missing !

- In line 352, the style is not uniform !

- In line 353, the style is not uniform ! Abbreviation is missing !

I remain open to any further clarification possibly needed from the Editorial Office.

To conclude, I Recommend Consideration Under Major Revision.

Please also note the supplement to this comment:
https://www.nonlin-processes-geophys-discuss.net/npg-2019-22/npg-2019-22-RC1-supplement.pdf

---

## Author Comment (AC2) · 3 Aug 2019

**Response to Reviewer 1:**

We are very grateful to your comments for the manuscript. They have important guiding significance for our manuscript and our research work. We have revised the manuscript according to your comments. The response to each revision is listed as follows:

*Comment 1*

The method of detection of anomalies of the borehole strain, is not well-known to non-specialists, and – I would say – to the specialists neither. In my opinion, at least one additional explanatory paragraph entirely devoted to this subject is needed, in the "Introduction" Section.

***Response:***

This is a constructive suggestion! We did not mention the background of negentropy in the "Introduction" Section. An explanatory paragraph has been supplemented. The corresponding references are also added to the "References" Section.

***Changes:***

We have supplemented an explanatory paragraph after Line 49 in the "Introduction" Section:

"Hence, it is implied that possible precursor anomalies lead to an increase in disordered components of observation data during earthquake preparation processes. K. Eftaxias et al. (2008) proved that the pre-catastrophic stage could break the persistency and high organization of the electromagetic field through studying fractional-Brownian-motion-type model using laboratory and field experimental electromagetic data. In view of Lévy flight and Gaussian processes, Lévy flight mechanism prevents the organization of the critical state to be completed before earthquakes, since the long scales are cut-off due to the Gaussian mechanism (S.M. Potirakis et al., 2019).

Entropy can serve as a measure of the unknown external energy flow into the seismic system (Akopian, S. T., 2014). K. Karamanos et al. (2006, 2005) quantified and visualized temporal changes of the complexity by approximate entropy, they claimed significant complexity decrease and accession at the tail of the preseismic electromagetic emission could be diagnostic tools for the impending earthquake. Yukio Ohsawa (2018) detected earthquake activation precursors by studying the regional seismic information entropy on earthquake catalog. Angelo De Santis (2011) recalled the Gutenberg

- Richter law and considered the negative logarithm of b-value is the entropy of the magnitude frequency of earthquake occurrence associated with two earthquakes in Italy.

Negentropy definition is based on the entropy and it is also widely used to detect non-Gaussian features. Yue Li (2018) proposed an arrival-time picking method based on negentropy for microseismic data. In this study, the negentropy is applied to borehole strain at Guza station associated with the Wenchuan earthquake, approximated by skewness and kurtosis. Subsequently we study the extracted negentropy anomalies in different scales to investigate correlations with crustal deformation."

**Comment 2**

In my opinion, in Fig. 6, "$kurtosis = 0.28699skewness^2 - 0.28696$" should boil down to "$kurtosis = 0.287(skewness^2 - 1)$", I mean that in equation (9), A=B which is a Remarkable result, if it holds true !!! At least one additional explanatory paragraph entirely devoted to this result is needed, in the "Discussion and Conclusions" Section !

**Response:**

We can change the parabolic relation into "$kurtosis = 0.287(skewness^2 - 1)$". Since in our case, A is always equal to -B because the kurtosis and skewness of the study period are normalized. There are

$$E(skewness) = E(kurtosis) = 0 \tag{1}$$

and

$$D(skewness) = E(skewness^2) = D(kurtosis) = E(kurtosis^2) = 1 \tag{2}$$

According to equation (9) in the manuscript and equation (1) and (2), there is

$$E(kurtosis) = E(A \cdot skewness^2 + B)$$

$$= \frac{1}{n}\sum_{i=1}^{n} A \cdot skewness^2(i) + n \cdot B$$

$$= A \cdot \frac{1}{n}\sum_{i=1}^{n} skewness^2(i) + B$$

$$= A \cdot D(skewness) + B$$

$$= A + B = 0 \qquad (3)$$

Thanks to your inspiration, we have derived a new relation based on this relation and supplemented it after equation (9) in the "Methodology" Section. Besides, the corresponding explanation has been supplemented in "Discussion and Conclusions" Section.

*Changes:*

We have supplemented an additional equation after equation (9) in Line 118-120 in the "Methodology" Section:

"Here we calculate the normalized skewness and kurtosis in the study period, so equation (9) can be derived into

$$kurtosis(X) = A \cdot (skewness^2(X) - 1) \qquad (10)$$

indicating the test day is super-Gaussian when the skewness is outside the range (-1,1)."

We have also supplemented an explanatory paragraph after Line 239 in the "Discussion and Conclusions" Section:

"In the skewness-kurtosis domain, we observed the evolution of the negentropy distribution prior to the earthquake. Negentropy gradually transformed its distribution to a parabolic relation since July 2007, indicating a relatively stable state was broken due to the non-Gaussian mechanism ."

**Comment 3**

In line 153, is stated that "k*=1.1130". What is the meaning of keeping so many significant digits ? Why not " k*=1.1 " or "k*=1.11" ? Please explain ! At least one additional explanatory paragraph is needed !

**Response:**

The meaning of k* value itself is a threshold for extracting negentropy anomalies. First, k* is calculated by Otsu's method by searching for k when the within-class variance of negentropy becomes the maximum, according to equations (10) to (13). Second, the format of the negentropy depends on the sample data. The YRY-4 borehole strainmeter has a measurement accuracy of $10^{-9}$, so we usually cutoff four digits after the decimal point in practical calculations. Then, the calculated k* is consistent with the accuracy of the negentropy and the strain data, resulting in 5 significant digits.

In fact, when we take the threshold k* as 1.1 or 1.11, there are 367 or 363 anomaly days respectively in study period (912 days), which is almost no difference with "k*=1.1130" (363 anomaly days). However, we still keep this result for the above reason when the numbers of anomalies are counted and accumulated.

**Changes:**

We have supplemented an explanatory paragraph after Line 155:

"Otsu threshold k* here is consistent with the accuracy of the negentropy and the strain data, The YRY-4 borehole strainmeter has a measurement accuracy of $10^{-9}$, however, we usually cutoff four digits after the decimal point in practical calculations."

**Comment 4**

In line 157, Fig.5, explain the Units !

***Response:***

X-axis and y-axis of the Fig. 5 are negentropy and its variance. The negentropy is defined as the weighted square of skewness and kurtosis in equation (6) in this manuscript. Because the skewness and kurtosis can be seen as ratios according to equation (7) and (8), there are no units.

***Minor corrections:***

Thanks for helping us with these typing errors.

- In line 17, "earthqake" → "earthquake" !

It has been modified.

- In lines 26 and 27, the citation has no uniform style !

(M.J.S. Johnston et al., 2006, Chi S. L. et al., 2014) has been modified as (Johnston M.J.S. et al., 2006, Chi S. L. et al., 2014).

- In lines 295 and 296, of the References list, there are quotation marks in the title of the Reference. This is the only place in the whole list where this happens !

It has been modified.

- In line 153, there are superscripts in the middle of the sentence, for no reason !

It has been modified.

- In line 157, the end dot (final punctuation mark) is missing !

It has been added.

- In line 159, it is mentioned "Fig 6(a)" instead of the correct "Fig. 6(a)" (the dot is missing) !

It has been added.

- In line 297, "Gutenber" → "Gutenberg" !

It has been modified.

- In line 325, the style is not uniform ! Dots are missing !

It has been modified.

[revised manuscript text omitted]

In the skewness-kurtosis domain, we observed the evolution of the negentropy distribution prior to the earthquake. Negentropy gradually transformed its distribution to a parabolic relation since July 2007, indicating a relatively stable state was broken due to the non-Gaussian mechanism .

Previous studies for the Wenchuan earthquake are consistent with our findings. Wang (2018) concluded that an apparent stress change occurred after June 2007 based on multiple focal mechanisms. Likewise, we did not find negentropy anomalies in the first six months of 2007. In the large-scale analysis, we show that the cumulative frequency of negentropy anomalies follows the a power-law behaviour approaching a critical time that is close to the earthquake time, and then recovers as a typical recovery phase after the earthquake This process is consistent with the empirical phenomena before and after earthquakes, which is also similar with a potential earthquake precursory pattern in magnetic data from Swarm satellites by A. De Santis (2017) for the 2015 Nepal event.

[revised manuscript text omitted]

---

## Referee Comment (RC2) · Anonymous Referee #2 · 14 Aug 2019

Review of the manuscript entitled: "Negentropy anomaly analysis of the borehole strain associated with the Ms 8.0 Wenchuan earthquake" by Zhu et al.

I reviewed this manuscript based on the author's responses to reviewer 1, consequently I did not comment the statistical method (for which I'm not expert), which part has been answered by the authors.

The authors analyze the strain data recorded by a borehole strainmeter (GUZA) distant from the epicenter of the 2008 Wenchuan earthquake by about 150 km. They use negentropy to investigate possible precursor strain anomalies which they attempt to relate to the earthquake nucleation process. The manuscript is mostly well organized and

written, the figures are mostly of sufficient quality excepted Fig.1 which is too dark and lack details about the active faults and the earthquake rupture location. The method is interesting and the crustal strain changes highlighted are intriguing. However, the authors are too optimistic about their results and should discuss their findings more carefully.

1. Major comments :

1 - In particular, the nearbiest strain station (GUZA) is located far from the earthquake source (150 km) , so that the network configuration to study strain precursors is far to be optimal. If the precursory phase implied widespread crustal changes, some changes should have been detected by other sensors, therefore other set of data (GPS ? Seismometers ? Groundwater ? ...) located in the near-field of the earthquake should be analyzed. Despite strainmeters are highly sensitive instruments, I have concerns about their capability to detect subtle strain changes at such large distance. Strain signals are mostly sensitive to local variations (hydrology, rain, air pressure, ...), so it would be interesting to see precipitation, groundwater and barometric records near GUZA station if available. I agree that the observed strain changes are spurious and the fact that they may roughly coincide with the onset time of the event makes them even more interesting, but there is absolutely no evidence that they are linked to the precursory phase of the earthquake. If the negentropy increased before the earthquake,why did it stay to a high level months after the rupture (Fig. 4) ?

In particular at L. 305-307 (and also L. 296-299), the authors stated that Âń negentropy anomalies ... may be a reflection of the subsurface medium and fault activities in the focal area associated with the Wenchuan earthquake Âż. This is a strong conclusion which came with no proof. Thus, based on only one station, the authors should point out that some strain changes are spurious but they shouldn't try to link these changes to the precursory phase with such a few observations. Therefore, the Discussion section should be modified and it should be clearly stated that further data are required to decipher a potential precursory phase.

2- It's not clear which data the authors use for the statistical analysis. Equations (1) and (2) describe the protocol to derive areal strain from borehole gauge measurements and they show that the 3 ways provide roughly similar areal strain signal. However, in L. 91-92, the authors calculate the difference in the data. Why that ? And what does this sentence (L. 91-92) means ? Is it the difference in strain data which is used for negentropy analysis ? If yes, why not using directly the areal strain signals which are a robust measure of crustal strain changes ? The authors removed tidal strain, but what about borehole trend and air pressure correction ? The description of the data is confusing and should be improved.

2. Other comments :

- Abstract (L. 7) : 12 May 2012 → 12 May 2008.

- Introduction : L. 22-27 is confusing as it gives the impression that precursory strain has been detected prior to the 2013 Ruisui earthquake (Canitano et al., 2015), which has not.

Besides, as the study involves the use of strain signals to study preseismic changes, it would be interesting to have exemples of previous studies which aimed to detect changes in the hypocentral regions of large earthquakes using strain data. For instance, short- period strain observations prior to the 1987 Supersition Hills earthquake (Agnew & Wyatt, 1989), 1989 Loma Prieta EQ (Johnston et al., 1990), 2010 L'Aquila (Amoruso & Crescentini, 2010) or 2013 Ruisui (Canitano et al., 2015) were all unsuccesfull. Note that those studies have been conducted on several stations located in the near-field of the shock, therefore under more optimal detection conditions.

- L. 25 : 'borehole strain data, which record the direct crustal changes' → borehole strainmeters which detect the crustal changes. Why 'direct' crustal changes ?

- L. 43 : non-Gaussian → non-Gaussian distribution.

- L. 57-58 : Âń Hence, it is implied ... preparation processes Âż : do you have a

reference for this sentence ?

- L. 65-66 : 'dozens of meters' : can you be more specific ?

- L. 178-180 : it is not clear why negentropy anomalies clustered on the left side of the parabola could be a signature of crustal deformation related to earthquake ?

- L. 200 : please consider remove 'famous'.

- L. 237-238 : the authors stated that anomalies increased in 2008 when earthquake aproaches and decreased after. That's no so obvious according to Fig. 4 for which anomaly rate seems to increase after the earthquake. Can you explain why ?

- L. 244-245 : Can you explain how you link the earthquake moment with the estimate of the inflection point based on negentropy analysis ? What does that mean that the earthquake moment is proved to be a critical time during the earthquake ?

- Fig. 8 : where is the critical point ? Can you explain it further ?

---

## Referee Comment (RC3) · Anonymous Referee #1 · 25 Aug 2019

Comments by Referee#1 concerning NPG submission:

npg-2019-22

Title: Negentropy anomaly analysis of the borehole strain associated with the Ms 8.0 Wenchuan earthquake Author(s): Kaiguang Zhu et al.

General Comments:

All the Authors' corrections and replies are in the right direction.

- Probably there is a misunderstanding concerning my first comment. I did not propose a whole new paragraph about the "Negentropy" notion but about the "borehole

strain" method ! However it has been a good idea to add a new paragraph about the "Negentropy" notion !

- The NPG paper by Karamanos et al. (NPG, 2005) mainly uses the "block-entropy analysis by lumping" of symbolic sequences, formally introduced in the literature in:

"Entropy analysis of substitutive sequences revisited." K. Karamanos J. Phys. A: Math. Gen. 34, 9231-9241 (2001).

- The Phys. Rev. E paper by Karamanos et al. (Phys. Rev. E, 2006) mainly uses the "T-complexity" notion to preseismic precursors for the first time in the international literature of Geophysics.

- The notion of the approximate Entropy (ApEn) is mainly used in:

"A unified approach of catastrophic events." S. Nikolopoulos, P. Kapiris, K. Karamanos and K. Eftaxias Nat. Haz. Earth Syst. Sciences 4, 615-631 (2004).

Of course, I leave to the discretion of the Editor and the Authors to add or not the above mentioned References.

I have no further comments.

Please also note the supplement to this comment:
https://www.nonlin-processes-geophys-discuss.net/npg-2019-22/npg-2019-22-RC3-supplement.pdf

---

## Author Comment (AC3) · 29 Aug 2019

Thank you for your suggestion and the references provided, we will make good use of them. We consider to add the references in the final version after discussing with the editor.
* * *

---

## Author Comment (AC4) · 11 Sep 2019

We are very grateful to your comments for the manuscript. We are uploading (a) Response to reviewer 2 and (b) the Revised manuscript with red label indicating changes. Please find them in the Supplement.

Please also note the supplement to this comment: https://www.nonlin-processes-geophys-discuss.net/npg-2019-22/npg-2019-22-AC4-supplement.zip

2019-22, 2019.

---

## Referee Report (RR1)

Review of the revised manuscript entitled: **"Negentropy anomaly analysis of the borehole strain associated with the Ms 8.0 Wenchuan earthquake"** by Zhu et al.

The authors have done a good job in responding the comments and modifying the manuscript. By adding Sec.5 where they compare their findings for further sensors and other periods (non disturbed by large shocks), they provide further evidence to strenghten the unusual character of their observations. They also ruled out a possible influence of environmental factors on the strain signals during the 2008 Wenchuan earthquake. I agree with their conclusion which stated that although the negentropy method and its results are interesting (and intriguing), linking the observed changes to the nucleation process still requires further data and analyses.I therefore recommend publication in NPG journal.

One point can still be improved concerning the description of the data (Sec. 2) and particularly how the authors derive the 'residual high frequency signals' (in Fig. 3). The authors explained clearly their approach in the response letter to reviewer 2 (as steps 1 & 2, pp. 9-10), also it would be interesting if they can also detailed the protocol (basically steps 1 & 2) in the manuscript.

Minor comments :

- Abstract (L. 2) : you can say that you analyze negentropy for 3 strainmeters (not only GUZA) ?

- Abstract (L. 7) : 'Combined with the confusion discussion …' : what you refer here is not clear, maybe just provide major results of this section in the Abstract ?

- Introduction (L. 10) : 'at least some large' → at least **for** some large.

- Introduction (L. 13) : Hsu et al. (2015) paper does not analyze earthquake strain signals (only meteorological perturbations), there may be a confusion here ?

- Introduction (L. 50) : the authors may consider changing the name of the 'Confusion Discussion' (Sec. 5), that sounds awkward when reading the main text.

- Fig. 1 (legend) : please consider adding details (e.g., the blue rectangles show the strainmeter stations, the epicenter is shown by yellow star + add the timing of the mainshock, 12 May 2008).

- L. 184-185 : 'Thus, we consider the Wenchuan earthquake day may be a critical time during the whole Wenchuan earthquake process' → that sentence is weird and has no clear purpose, you may consider remove it.

L. 196-199 : these sentences are confusing, please consider rewrite them.

L. 218-219 : 'However, since the curve is approximately, … the value of inflection point exists a range.' → which range ? Something is missing in this sentence.

L. 219 : may **be** able.

L. 219 and 224 : 'receive' → rather 'detect' or 'record'.

L. 243 :'Through the confusion discussion' → modify by 'In Sec. 5' for exemple.

---

## Author Response (AR3)

**We are uploading our point-by-point response to all referee comments and specify all changes in the revised manuscript, combined with a marked-up manuscript version showing the changes.**

**Response to Reviewer 1:**

We are very grateful to your comments for the manuscript. They have important guiding significance for our manuscript and our research work. We have revised the manuscript according to your comments. The response to each revision is listed as follows:

*Comment 1:*

The method of detection of anomalies of the borehole strain, is not well-known to non-specialists, and – I would say – to the specialists neither. In my opinion, at least one additional explanatory paragraph entirely devoted to this subject is needed, in the "Introduction" Section.

*Response:*

This is a constructive suggestion! We did not mention the background of negentropy in the "Introduction" Section. An explanatory paragraph has been supplemented. The corresponding references are also added to the "References" Section.

*Changes:*

We have supplemented an explanatory paragraph in the "Introduction" Section:

"Hence, it is implied that possible precursor anomalies lead to an increase in disordered components of observation data during earthquake preparation processes. K. Eftaxias et al. (2008) proved that the pre-catastrophic stage could break the persistency and high organization of the electromagetic field through studying fractional-Brownian-motion-type model using laboratory and field experimental electromagetic data. In view of Lévy flight and Gaussian processes, Lévy flight mechanism prevents the organization of the critical state to be completed, since the long scales are cut-off due to the Gaussian mechanism (S.M. Potirakis et al., 2019).

Entropy can serve as a measure of the unknown external energy flow into the seismic system (Akopian, S. T., 2014). K. Karamanos et al (2005, 2006) quantified and visualized temporal changes of the complexity by approximate entropy, they claimed significant complexity decrease and accession at the tail of the preseismic electromagetic emission could be diagnostic tools for the impending earthquake. Yukio Ohsawa (2018) detected earthquake activation precursors by studying the regional seismic information entropy on earthquake catalog. Angelo De Santis (2011) recalled the Gutenberg - Richter law and considered the negative logarithm of b-value is the entropy of the magnitude frequency of earthquake occurrence associated with two earthquakes in Italy.

Negentropy definition is based on the entropy and it is also widely used to detect non-Gaussian features. Yue Li (2018) proposed an arrival-time picking method based

on negentropy for microseismic data. In this study, the negentropy is applied to borehole strain at Guza station associated with the Wenchuan earthquake, approximated by skewness and kurtosis. Subsequently we study the extracted negentropy anomalies in different scales to investigate correlations with crustal deformation."

**Comment 2:**

In my opinion, in Fig. 6, "kurtosis=0.28699skewness^2-0.28696" should boil down to "kurtosis=0.287(skewness^2-1)", I mean that in equation (9), A=B which is a Remarkable result, if it holds true !!! At least one additional explanatory paragraph entirely devoted to this result is needed, in the "Discussion and Conclusions" Section !

*Response:*

We can change the parabolic relation into "kurtosis=0.287(skewness^2-1)". Since in our case, A is always equal to -B because the kurtosis and skewness of the study period were normalized. There are

$$E(skewness) = E(kurtosis) = 0, \tag{1}$$

and

$$D(kurtosis) = E(kurtosis^2) = D(skewness) = E(skewness^2) = 1. \tag{2}$$

According to equation (9) in the manuscript and equation (1) and (2), there is

$$\begin{aligned} E(kurtosis) = E(A \cdot skewness^2 + B) &= \frac{1}{n}\sum_{i=1}^{n}(A \cdot skewness^2(i) + n \cdot B) \\ &= A \cdot \frac{1}{n}\sum_{i}^{n} skewness^2(i) + B \\ &= A \cdot D(skewness) + B \\ &= A + B = 0 \end{aligned} \tag{3}$$

Thanks to your inspiration, we have derived a new relation based on this relation and supplemented it after equation (9) in the "Methodology" Section. Besides, the corresponding explanation has been supplemented in "Discussion and Conclusions" Section.

*Changes:*

We have supplemented an addition equation after equation (9) in Line 101-103 in the "Methodology" Section:

"Here we calculate the normalized skewness and kurtosis in the study period, so equation (9) can be derived into

$$kurtosis(X) = A \cdot (skewness^2(X) - 1) \tag{10}$$

indicating when the negentropy is outside the range (-1,1), the test day is super-Gaussian."

We have also supplemented an explanatory paragraph. But combined with the comments of reviewer 2, the explanatory paragraph was not shown in the final revised

manuscript.

"In the skewness-kurtosis domain, we observed the evolution of the negentropy

distribution prior to the earthquake. Negentropy gradually transformed its distribution
to a parabolic relation since July 2007, indicating a relatively stable state was broken

due to the non-Gaussian mechanism ."

**Comment 3:**
In line 153, is stated that "k*=1.1130". What is the meaning of keeping so many significant digits ? Why not " k*=1.1 " or "k*=1.11" ? Please explain ! At least one additional explanatory paragraph is needed !

*Response:*
The meaning of k* value itself is a threshold for extracting negentropy anomalies. First, k* is calculated by Otsu's method by searching for k when the within-class variance of negentropy becomes the maximum, according to equations (10) to (13). Second, the format of the negentropy depends on the sample data. The YRY-4 borehole strainmeter has a measurement accuracy of $10^{-9}$, but in practical calculations, we usually cutoff four digits after the decimal point. Therefore, the calculated k* is consistent with the accuracy of the negentropy and the strain data, resulting in 5 significant digits.
In fact, when we take the threshold k* as 1.1 or 1.11, there are 367 or 363 anomaly days respectively in study period (912 days), which is almost no difference with "k*=1.1130" (363 anomaly days). However, we still keep this result for the above reason when the numbers of anomalies are counted and accumulated.

*Changes:*

We have supplemented an explanatory paragraph after Line 131:
"Otsu threshold k* here is consistent with the accuracy of the negentropy and the strain data, The YRY-4 borehole strainmeter has a measurement accuracy of $10^{-9}$, however, we usually cutoff four digits after the decimal point in practical calculations."

**Comment 4:**
In line 157, Fig.5, explain the Units !

*Response:*
X-axis and y-axis of the Fig. 5 are negentropy and its variance. The negentropy is defined as the weighted square of skewness and kurtosis in equation (6) in this manuscript. Because the skewness and kurtosis can be seen as ratios according to equation (7) and (8), there are no units.

**Minor corrections:**

(1)  - In line 17, "earthqake" → "earthquake" !
It has been modified.
(2)  - In lines 26 and 27, the citation has no uniform style !
(M.J.S. Johnston et al., 2006, Chi S. L. et al., 2014) has been modified as
(Johnston M.J.S. et al., 2006, Chi S. L. et al., 2014).
(3)  - In lines 295 and 296, of the References list, there are quotation marks in the title
of the Reference. This is the only place in the whole list where this happens !
It has been modified.
(4)  - In line 153, there are superscripts in the middle of the sentence, for no reason !
It has been modified.
(5)  - In line 157, the end dot (final punctuation mark) is missing !
It has been added.
(6)  - In line 159, it is mentioned "Fig 6(a)" instead of the correct "Fig. 6(a)" (the dot
is missing) !
It has been added.
(7)  - In line 297, "Gutenber" → "Gutenberg" !
It has been modified.
(8)  - In line 325, the style is not uniform ! Dots are missing !
It has been modified.

**References:**

K. Karamanos, A. Peratzakis, P. Kapiris, S. Nikolopoulos, J. Kopanas and K. Eftaxias. Extracting preseismic electromagnetic signatures in terms of symbolic dynamics. Nonlinear Processes in Geophysics 12, 835-848, 2005.

K. Karamanos, D. Dakopoulos, K. Aloupis, A. Peratzakis, L. Athanasopoulou, S. Nikolopoulos, P. Kapiris and K. Eftaxias. Study of pre-seismic electromagnetic signals in terms of complexity. Phys. Rev.E 74, 016104 – 016125, 2006.

K. Eftaxias, Y. Contoyiannis, G. Balasis, K. Karamanos, J. Kopanas, G. Antonopoulos, G. Koulouras and C. Nomicos. Evidence of fractional-Brownian-motion-type asperity model for earthquake generation in candidate pre-seismic electromagnetic emissions. Nat. Haz. Earth Syst. Sci. 8, 657-669, 2008.

S. M. Potirakis, Y. Contoyiannis and K. Eftaxias. Levy and Gauss statistics in the preparation of an earthquake. Physica A, Vol. 528, 15 August 2019, 121360 (In Press)

De Santis, A., et al.: The Gutenberg-Richter Law and Entropy of Earthquakes: Two Case Studies in Central Italy. Bulletin of the Seismological Society of America 101(3): 1386-1395, 2011.

Li, Y., et al.: Arrival-time picking method based on approximate negentropy for microseismic data. Journal of Applied Geophysics 152: 100-109, 2018.

Ohsawa, Y.: Regional Seismic Information Entropy to Detect Earthquake Activation Precursors. Entropy 20(11), 2018.

Akopian, S. T.: Open dissipative seismic systems and ensembles of strong earthquakes: energy balance and entropy funnels. Geophysical Journal International 201(3):

1618-1641, 2015.

**Response to Reviewer 2:**

We are very grateful to your comments for the manuscript. They have important guiding significance for our manuscript and our research work. We have revised the manuscript according to your comments. The response to each revision is listed as follows:

*1. The manuscript is mostly well organized and written, the figures are mostly of sufficient quality excepted Fig.1 which is too dark and lack details about the active faults and the earthquake rupture location.*

**Response:**

Thank you for your suggestion. Fig.1 in the manuscript is not professional enough. We have updated the figure with the active faults and the earthquake rupture location. Besides, we have supplemented three stations aiming to discuss our findings in the "Confusion discussion" Section.

[Figure]

Fig. 1 Location map showing the epicentre of the Wenchuan earthquake and three stations. The epicentre was located in Wenchuan County, Sichuan Province, at 31.01°N, 103.42°E. The red circles are aftershocks (Ms>3.0) from the main shock to October 2008. The green curve is the schematic curve of the main rupture zone and the black curves are faults.

**2. Major comment 1:**
- *In particular, the nearest strain station (GUZA) is located far from the earthquake source (150 km), so that the network configuration to study strain precursors is far to*

*be optimal. If the precursory phase implied widespread crustal changes, some changes should have been detected by other sensors, therefore other set of data (GPS? Seismometers? Groundwater? ...) located in the near-field of the earthquake should be analyzed. Despite strainmeters are highly sensitive instruments, I have concerns about their capability to detect subtle strain changes at such large distance. Strain signals are mostly sensitive to local variations (hydrology, rain, air pressure, ...), so it would be interesting to see precipitation, groundwater and barometric records near GUZA station if available. I agree that the observed strain changes are spurious and the fact that they may roughly coincide with the onset time of the event makes them even more interesting, but there is absolutely no evidence that they are linked to the precursory phase of the earthquake. If the negentropy increased before the earthquake, why did it stay to a high level months after the rupture (Fig. 4)?*

*In particular at L. 305-307 (and also L. 296-299), the authors stated that ´n negentropy anomalies ... may be a reflection of the subsurface medium and fault activities in the focal area associated with the Wenchuan earthquake Â˙z. This is a strong conclusion which came with no proof. Thus, based on only one station, the authors should point out that some strain changes are spurious but they shouldn't try to link these changes to the precursory phase with such a few observations. Therefore, the Discussion section should be modified and it should be clearly stated that further data are required to decipher a potential precursory phase.*

**Response:**

Thank you very much for your suggestion.

We agree the "Discussion and Conclusion" section is kind of unreasonable based on the borehole strain data of one station.

Referring to your guiding suggestions above, we have studied the correlation between the negentropy anomalies and the Wenchuan earthquake through three parts. 1. Comparison of random time periods. 2. Comparison of different stations. 3. Exclusion of co-seismic events and weather factors.

We first introduce the three supplemented parts and answer your other specific questions next.

**Changes:**

We have divided the "Discussion and Conclusion" Section into "Confusion Discussion" Section and "Conclusion" Section.

We have also discussed our findings more carefully in the "Confusion Discussion" Section and objectively stated further researches are required to decipher a potential precursory phase in the "Conclusion" Section.

To study the correlation between extracted anomalies and earthquakes, Parrot (2011) proposed a method for random earthquake distribution. The location of the earthquake epicentre is randomly changed, but the size of the study area and the study time is kept unchanged. The anomalous variation is compared between the random region and the

actual seismic region, and the correction between the anomalies and the earthquake is determined. In order to verify the relationship between negentropy anomalies and the Wenchuan earthquake, we did a similar random earthquake distribution study as follows.

**1. Comparison of random time periods.**

We randomly selected March 20, 2011 and March 24, 2014 as the random earthquake days, and study the strain data for 200 days before and after the earthquake days at Guza station. The selected periods are required to be in the absence of strong earthquakes and with higher quality. We performed negentropy analysis on these two random periods and compared them with the results of negentropy analysis associate with the Wenchuan earthquake as shown in Fig. 2.

[Figure]

Fig. 2 The comparative analysis of cumulative frequency of negentropy anomalies between earthquake period and random time periods. The green triangles correspond to the random earthquake on March 20, 2011, the blue triangles correspond to the random earthquake on March 24, 2014.

As we can see in Fig. 2, the cumulative frequency of negentropy anomalies of random periods have statistical linear increases. However, in the Wenchuan earthquake periods, as the earthquake approaches, the cumulative frequency of negentropy anomalies increases rapidly and recovered to a slow growth after the earthquake.

**2. Comparison of different stations.**

We supplemented two other stations to find out if their observations received strain changes. We chose Xiaomiao station and Renhe station, their locations are shown in Fig. 1. Corresponding to the Guza station, we did the negentropy analysis of the two stations as shown in Fig. 3.

[Figure]

Fig. 3 Cumulative frequency of negentropy anomalies of Xiaomiao station and Renhe station from September 16, 2007 to June 30, 2009. The negentropy analysis of Guza station is from January 1, 2007 to June 30, 2009, because of the different installation time of the instruments. The red vertical line is the inflection point of the fitting curve of Guza station. The blue vertical line is the inflection point of the fitting curve of Xiaomiao station. The black dashed line is the earthquake day.

As we can see in the Fig. 3, the cumulative frequency of negentropy anomalies of Xiaomiao station are also well fitted by the sigmoid function. The accumulation curve is growing rapidly before the earthquake and concave downward after which is similar to the Guza station, although the inflection point of Xiaomiao station is about two months preceding the earthquake moment. However, since the curve is approximately linear before and after the inflection point, the value of inflection point exists a range.

Cumulative anomalies of the Renhe station are basically linear, indicating that the Renhe stations may don't receive pre-earthquake anomalies.

Renhe station is far from the end of the Wenchuan earthquake fault, according to the fracture mechanics, so it is reasonable that no abnormal changes are observed. However, Xiaomiao station is located between the Guza station and Renhe station, it may receive some changes. The fitting result also shows that there is a similar trend to the Guza station, with a weaker curvature. So, for the nearest station to the epicentre, Guza station may able to receive more pre-earthquake anomalies.

Furthermore, Qiu (2012) found that the anomalies at Ningshan station were similar to the anomalies at Guza station. Such two stations have observed similar Wenchuan earthquake precursor anomalies, which may not be accidental. Since the Ningshan station is actually located at the northeastern end of the Longmenshan fault zone. This location is a correspondence with the southwestern end of the fault where the Guza station is.

**3. Exclusion of co-seismic events and weather factors.**

We studied the Earthquake events data from the USGS National Earthquake Information Center (NEIC) catalog instead of continuous seismic waveforms recorded by seismometers, and consider all the Ms3.0+ events in the catalog which occurred in the study region during the study period as shown in Fig. 4(a). We count the multiple earthquakes occurred in one day as one event. Comparing the results in the manuscript,

we accumulate the earthquake event as Fig. 4(b). Before the earthquake, the cumulative frequency of earthquake events increased linearly, indicating there was no rapid growth phase of earthquake events in the region. This phenomenon is different from the cumulative frequency of negentropy anomalies extracted by borehole strain, which also verify that the co-seismic events didn't cause the pre-earthquake anomalies recorded by borehole strain before the Wenchuan earthquake at Guza station. While after the Wenchuan earthquake, there is a rapid growth rate of the cumulative frequency due to numerous aftershocks. With the restoration of the crust in the seismic source region, the accumulation after the Wenchuan earthquake is gradually slow, which is similar to the accumulation of negentropy anomalies after the earthquake at Guza station.

[Figure]

Fig. 4 Earthquake catalog (Ms>3.0) of the study region during the study period and the cumulative frequency of earthquake events.

We also agree that strain signals are mostly sensitive to a few meteorological factors, such as the air pressure, temperature and rainfall. The water level data of Guza station are not available before 2009. As shown in Fig. 5, we display the detrend borehole strain, pressure variations, temperature variations recorded at Guza station and the daily rainfall measured by Tropical Rainfall Measuring Mission (TRMM) satellite which are downloaded through the NASA GIOVANNI-4 for the same period and the same area (http://giovanni.gsfc.nasa.gov/giovanni/) . There are clearly annual variations in the strain, air pressure, temperature and rainfall data. The air pressure and temperature have been steadily fluctuating within a certain range, and the rainfall is also shown to be more in summer and less in winter.

While we calculated the differential data of the strain for negentropy analysis. So, we make differential calculations for all three influencing factors as shown in Fig. 6.

We observed that the air pressure, temperature and rainfall didn't change abnormally during the period when the extracted anomalies increase, whether we do differential calculation.

To further examine the correlation, we calculated the correlation coefficient in each sliding window (win=15 days) between the factors and the strain, and the results are shown in the Fig. 7. Although the original factors and the original strain are not strongly corrected, the correlation coefficients of differential data are far less than those of the

original data. Therefore, we consider that the abnormal variations on the processed strain signals are not caused by these factors.

Thanks for your suggestion, we revisited the strain for different time periods, different stations and factors, then discussed them carefully. Considering the structure of the manuscript, for the meteorological factors, we have only briefly discussed and excluded them. A revised manuscript was attached in the updated manuscript.

[Figure]

Fig. 5. Borehole stain, air pressure, temperature and rainfall variations during study period at Guza station

[Figure]

Fig. 6. Differential borehole stain, air pressure, temperature and rainfall variations during study period at Guza station

[Figure]

Fig. 7 (a), (b) and (c) are the results of correction coefficient between air pressure, temperature, rainfall and strain. (d), (e) and (f) are the results of correction coefficient between differential air pressure, differential temperature, differential rainfall and differential strain.

**We detailed the method of data pre-processing in major comment 2.**

*# [1] In particular, the nearest strain station (GUZA) is located far from the earthquake source (150 km), so that the network configuration to study strain precursors is far to be optimal.*

**Response:**

As for the earthquake-monitoring capability of the borehole strain, Su Kaizhi (1991) comprehensively considered the time characteristics of the strainmeters detection capability, the amplitude distribution of the strain precursor and the occurrence time of the strain precursor, started with the general relationship between the time scale and the detected minimum strain amplitude, the epicenter distance and the magnitude, and magnitude and the occurrence time of the strain precursors, combined with observations of Chinese borehole strainmeters, and therefore obtained the estimated monitoring and controlling extent of long- and medium-term, short-term, and impending precursors. For the long-term precursor phase of the earthquakes of Ms8.0, the borehole strainmeter has a monitoring capability radius of about 430 km, and for the short-term and impending precursors, the scope is more than 700 km. So, the Wenchuan earthquake source is within the monitoring capability of the borehole strainmeter at Guza station.

Besides, in the Fig. 1, we find the Guza station stands on the southwestern end of the Longmenshan fault zone. From the point of view of fracture mechanics, the end point of the fracture is where the stress concentrated or even the singularity occurs (Qiu Z. H., 2012). According to this view, we think that the strainmeter is possible to record information related to earthquakes at Guza Station.

**Changes**:

We have supplemented a short explanation for this distance in Line 56-58.

*# [2] If the precursory phase implied widespread crustal changes, some changes should have been detected by other sensors, therefore other set of data (GPS? Seis- mometers?*

*Groundwater? ...) located in the near-field of the earthquake should be analyzed.*

**Response:**

We agree that it would be better to analyze other sets of data such as GPS and seismometers which may record some changes. Unfortunately, some data are limited for us for now. However, we can provide a few explanations.

Firstly, because borehole strainmeters are designed to record deformation that lies between the spectral coverage of seismometers and GPS, and are ideal for capturing strain transients that occur in periods of hours to years as shown in Fig. 8, we choose borehole strain to study the pre-earthquake changes.

[Figure]

Fig. 8 shows Instrumental characteristics of seismology, borehole tensor strain and GPS tectonics. Source: http://www.earthscope.org. Borehole strain are ideal for revealing short-term (from seconds to years); Seismographs are mainly used to determine relevant seismic parameters; GPS is a relatively long-term observation (from weeks to decades).

Secondly, we studied the Earthquake events data from the USGS NEIC catalog instead of continuous seismic waveforms recorded by seismometers. The comparison result is in Fig. 4.

However, different instruments record signals of different characteristics. We are not sure about the possibility that the crustal changes are recorded by other sensors. Besides, we investigated the outgoing longwave radiation (Kong X., et al., 2018), radon concentrations in water (Yan R., et al., 2011), water level and water temperature data (Sun X. L., et al., 2016) before the earthquake. There are some similarities between their results and the results in our manuscript. (We quoted their results at the end of this response.)

Thirdly, according to the principle of the YRY-4 strainmeter, the four observations $S_i$, $(i=1,2,3,4)$ are sampled by four independent sensors. This means that the sensors in the four directions do not affect each other. We can think that the anomalies before and after the Wenchuan earthquake were recorded simultaneously by four sensors.

**Changes:**

We have stated the further data and further researches are needed to confirm a potential precursor phase in "Conclusion" Section.

**Response:**

This is the empirical phenomenon after the main shock in the hypocentral region. The Wenchuan earthquake is main-aftershock type. After the Wenchuan earthquake, the earth's crust was still in a very unstable stage. As of October 2008, more than 900 aftershocks (Ms>3.0) occurred as shown in the Fig. 1. Therefore, the negentropy stay to a high level after the rupture.

In addition, with the restoration of the crust in the seismic source region, the accumulation of negentropy anomalies after the earthquake is growing slower as shown in Fig. 8 in the manuscript.

*# [4] In particular at L. 305-307 (and also L. 296-299), the authors stated that Ân negentropy anomalies ... may be a reflection of the subsurface medium and fault activities in the focal area associated with the Wenchuan earthquake. This is a strong conclusion which came with no proof.*

**Response:**

Thank you for your suggestion. This conclusion is indeed too positive. The mechanism for these abnormal changes is needed to be discussed further. What we want to express is the extracted negentropy anomalies may be related to the Wenchuan earthquake.

We have modified the conclusions more prudently in "Conclusion" Section.

**Major comment 2:**

*It's not clear which data the authors use for the statistical analysis. Equations (1) and (2) describe the protocol to derive areal strain from borehole gauge measurements and they show that the 3 ways provide roughly similar areal strain signal. However, in L. 91-92, the authors calculate the difference in the data. Why that? And what does this sentence (L. 91-92) means? Is it the difference in strain data which is used for negentropy analysis? If yes, why not using directly the areal strain signals which are a robust measure of crustal strain changes? The authors removed tidal strain, but what about borehole trend and air pressure correction? The description of the data is confusing and should be improved.*

**Response:**

Thank you for your suggestion. We did not describe the data pre-processing part clearly, especially in L. 91-92. The detailed process is as follows.

First, equations (1) and (2) in the manuscript describe the protocol to derive areal strain from borehole strainmeters. Then we show the component data satisfy self-consistent through Fig. 2 in the manuscript to illustrate that the areal strain can replace the observations of four components.

Second, we processed the areal strain. The procedures and reasons of data preprocessing are as follows.

**Step 1**: Differential calculation

We set the areal strain data as $X(n)$ and differential areal strain data as $Y(n)$, we know $Y(n) = X(n) - X(n-1)$, $F_X(e^{j\omega})$ is frequency characteristic of $X(n)$, $F_Y(e^{j\omega})$ is frequency characteristic of $Y(n)$ according to differential properties based on Fourier transform

$$F_Y(e^{j\omega}) = (1 - e^{-j\omega})F_X(e^{j\omega})$$

This process can be equivalent to a filtering system, let the frequency response of this system be $H_1(e^{j\omega})$, then

$$\left|H_1(e^{j\omega})\right| = \left|\frac{F_Y(e^{j\omega})}{F_X(e^{j\omega})}\right| = \left|1 - e^{-j\omega}\right|$$
$$= \sqrt{2(1 - \cos\omega)}$$

It can be seen that when $\omega$ is very small or 0, the frequency response is 0, indicating that the Step 1 removes the low frequency information of the signal, including borehole trend and low frequency effects of the air pressure and temperature on the signal.

**Step 2:** Harmonic analysis

We remove the periodic term that still exists through daily harmonic analysis. We set the fitting function as Fourier series. The reserved signal $Z(n)$ can be simplified as

$$Z(n) = Y(n) - \sum_{k=1}^{n} A_k \sin(k\omega_0 n + \varphi_i)$$

$F_Z(e^{j\omega})$ is frequency characteristic of $Z(n)$, this process can also be seen as a filtering system, let the frequency response of this system be $H_2(e^{j\omega})$.

$$\left|H_2(e^{j\omega})\right| = \left|\frac{F_Y(e^{j\omega}) - \sum_k \pi A_k [\delta(\omega - k\omega_0) - \delta(\omega + k\omega_0)]/2}{F_Y(e^{j\omega})}\right|$$

Minimize $Z(n)$ by least squares method in time domain, then ideally for the system gain

$$\left|H_2(e^{j\omega})\right| = \begin{cases} 0 & \omega = k\omega_0 \\ 1 & \omega = others \end{cases}$$

The frequency response after two steps is

$$|H(e^{j\omega})|=|H_1(e^{j\omega})H_2(e^{j\omega})|=\begin{cases} 0 & \omega=k\omega_0 \\ \sqrt{2(1-\cos\omega)} & \omega=others \end{cases}$$

Therefore, the Step 2 removes the periodic terms in the signal. We think the period terms here mainly includes the periods related to the solid tide, also includes the periodic effects of air pressure.

Finally, we performed the negentropy analysis for the processed data.

We randomly selected one day to explain the effects of the data pre-processing as shown in Fig. 9.

[Figure]

Fig. 9. (a). Areal strain of January 19, 2009. (c). Areal strain after differential calculation. (e). Differential areal strain after harmonic analysis. (b), (d), and (f) are the data distribution of (a), (c) and (c).

There is an abnormal change on the original areal strain curve at about 1400 minutes in Fig. 9(a). The areal strain is obviously non-Gaussian in Fig .9(b). The data presents a U-shaped distribution, and its negentropy has no meaning. This change becomes obvious after the differential calculation (Fig. 9(c)), but the negentropy is small due to the amplitudes of the periodic terms (Fig. 9(d)). After the harmonic analysis, the negentropy value increases significantly (Fig. 9(f)). Therefore, the small changes in the curve are amplified by the data pre-processing.

Our ultimate goal is to study negentropy (non-Gaussian characteristic) of the signals. The low frequency components and periodic components affect the Gaussian characteristic of the areal strain signals. This is why we processed the areal strain signals, although the areal strain is a robust measure of crustal strain changes. We are more concerned about the remaining high frequency variations. This is what we want to express in L. 91-92 in the manuscript.

Since the data processing part is not the most important part of this manuscript, we have not added all the description to the revised manuscript.

**Changes:**

We have updated the description of data processing in Line 64-71.

**Other comments :**
*- Abstract (L. 7) : 12 May 2012 → 12 May 2008.*
  We have modified.

*- Introduction : L. 22-27 is confusing as it gives the impression that precursory strain has been detected prior to the 2013 Ruisui earthquake (Canitano et al., 2015), which has not.*
*Besides, as the study involves the use of strain signals to study preseismic changes, it would be interesting to have exemples of previous studies which aimed to detect changes in the hypocentral regions of large earthquakes using strain data. For instance, short- period strain observations prior to the 1987 Supersition Hills earthquake (Agnew & Wyatt, 1989), 1989 Loma Prieta EQ (Johnston et al., 1990), 2010 L'Aquila (Amoruso & Crescentini, 2010) or 2013 Ruisui (Canitano et al., 2015) were all unsuccesfull. Note that those studies have been conducted on several stations located in the near-field of the shock, therefore under more optimal detection conditions.*

**Response:**
   Thank you for your suggestion. Canitano also gave us a short comment about his research. We have updated our expression.
   Besides, we have supplemented these references in "Introduction".

*- L. 25: 'borehole strain data, which record the direct crustal changes' →borehole strainmeters which detect the crustal changes. Why 'direct' crustal changes ?*

**Response:**
   Earthquake occurrence is the process of mechanics. "Direct" indicates that the borehole strain also record force, to distinguish from other kinds of observations. Strictly speaking, "direct" is a little inaccurate.
**Changes:**
   We have modified as "Borehole strainmeters which detect the crustal changes"

*- L. 43 : non-Gaussian → non-Gaussian distribution*
We have modified.

*- L. 57-58 : ´ n Hence, it is implied ... preparation processes Â˙ z : do you have a reference for this sentence ?*

**Response:**
   This is our language expression problem. In fact, this sentence follows the last two paragraphs. There are 3 references among them indicate this point of view, then, there are 2 other references in this paragraph also support this sentence.
**Changes:**
   We have modified this sentence into in Line 36. "Thereby, it is possible that precursor

anomalies lead to an increase of disordered components in observation data."

*-L. 65-66 : 'dozens of meters' : can you be more specific ?*
Yes, we have been more specific.
"… have been deployed at depths of more than 40 metres …"

*- L. 178-180 : it is not clear why negentropy anomalies clustered on the left side of the parabola could be a signature of crustal deformation related to earthquake ?*

**Response:**

There are two main reasons. First of all, these anomalies are different from this normal background. Since after the data processing, it is normal for the daily strain to be Gaussian distribution. Secondly, since the similar characteristics of the anomalies cause the cluster phenomenon to appear on the parabola before and after the Wenchuan earthquake.

Whether the negentropy falls on the left or right side of the parabola is determined by the sign of the skewness. The relationship between skewness and data distribution is as shown in Fig. 10. Negentropy anomalies (red points in Fig. 6 in the manuscript) clustered on the left side of the parabola shows that the observations of these abnormal days have negative skewness. In other words, there are a lot of negative extreme values in the observations of these days.

[Figure]

Fig. 10. Schematic diagram of the relationship between skewness and data distribution. Source: https://ss.csdn.net/p?https://mmbiz.qpic.cn/mmbiz_png/heS6wRSHVMkoXmWbecSLSvBtFqZRJ W9MPickoP99bO1zu6cbtBuI34xjKpOObcRGErLkeSVGRrToJgd8Cria3tqw/640?wx_fmt=png.

From Fig. 6 (b) in the manuscript, we can see as the earthquake approaches, the abnormal days are basically clustered on the left side of the parabola. According to W. Marzocchi et al. (2014), spatio-temporal clustering is generally believed to represent the most striking departure from randomness for the large earthquake occurrence process.

Therefore, we suspect negentropy anomalies clustered on the left side of the parabola could be related to the earthquake. In order to study the possible correspondence with the earthquake process, we calculate the cumulative frequency of negentropy anomalies later in the manuscript.

**Changes:**

We have updated this sentence as "Besides, the extracted negentropy anomalies are clustered strongly on the left side of the parabola, which exhibit similar characteristics different from the normal Gaussian distribution."

*- L. 200: please consider remove 'famous'.*
We have removed "famous".

*- L. 237-238: the authors stated that anomalies increased in 2008 when earthquake approaches and decreased after. That's no so obvious according to Fig. 4 for which anomaly rate seems to increase after the earthquake. Can you explain why?*

**Response:**
The anomalies there refer to the frequency of anomalies. In fact, the anomalies did not decrease immediately after the earthquake, since the earth's crust was still in a very unstable stage. As of October 2008, there are more than 900 aftershocks (Ms>3.0) as shown in the Fig. 1.

However, after extracting the negentropy anomalies that greater than the threshold in Fig. 4 in the manuscript, we calculated the cumulative frequency of negentropy anomalies and fit them by sigmoid function as shown in Fig. 8 in the manuscript. Based on the fitting result for the entire process in Fig. 8, we see the inflection point is almost at the earthquake moment. And before the inflection point, the accumulation of anomaly frequency has an exponential increase trend. After the inflection point, there is an opposite increase trend. Therefore, we stated that anomalies increased when earthquake approaches and decreased after.

*- L. 244-245: Can you explain how you link the earthquake moment with the estimate of the inflection point based on negentropy analysis? What does that mean that the earthquake moment is proved to be a critical time during the earthquake?*
*- Fig. 8: where is the critical point? Can you explain it further?*

**Response:**
We calculated the cumulative frequency of negentropy anomalies. Since in general, accumulated value of a typical random process usually has a linear increase. In particular, in case of critical phenomena, we would expect more frequent anomalies when they approach the critical point, and less frequent anomalies after (De Santis, A. et al. ,2017). However, the cumulative frequency of negentropy anomalies is well fitted by sigmoid function.

Sigmoid function is expressed as

$$y = A2 + \frac{(A1 - A2)}{(1 + e^{\frac{x - x0}{dx}})},$$

where $A1$, $A2$, $x0$ and $dx$ are calculated by fitting and $x0$ is the inflection point of the function. The sigmoid function is a power law temporal behavior with an upper concavity and a subsequent power-law behavior after the inflection, with an opposite

concavity.

Also, the value of $x_0$ obtained in the fitting result is very close to the time of the Wenchuan earthquake as shown in the Fig. 8 in the manuscript. The inflection point is the black vertical solid line, the earthquake day is the black dashed line. The two lines almost coincide in the Fig. 8 in the manuscript.

Therefore, we link the earthquake moment with the estimate of the inflection point.

Since the fitting curve is concave upward before the earthquake and concave downward after the earthquake. As the earthquake approaches, the slope of the curve increases, reaching its maximum near the earthquake, and then slowly decreasing.

Therefore, we stated the earthquake moment is proved to be a critical time during the earthquake. We also learned from De Santis, A. et al. (2017), they calculated the cumulative number of magnetic anomalies detected by Swarm satellite and also fit it by the sigmoid function. They thought inflection point in this function is a reasonable estimation of the time of the significant change in the critical dynamical system.

In Fig. 8, the critical point refers to the inflection point. We have unified the words as "infection point" and supplemented a brief explanation in Line 169-171 and Line 177-181.

We tried our best to improve the manuscript and made some revisions in the manuscript. These revisions will not influence the content and framework of the paper.

We appreciate for Editor and Reviewers' warm work earnestly, and hope that the correction will meet with approval.

**Once again, thank you for your advice, hope to be able to learn more knowledge from you.**

Appendix:

Kong X., et al. (2018) extracted the anomalies of the outgoing longwave radiation (OLR) data by calculating CD value (they defined).

[Figure]

Fig. 8. Comparative CD value of grids 1, 2, 3, 4 in Wenchuan area, from 28th September, 2007 to 12th May , 2008, yellow line represents Wenchuan earthquake date.

Sun X. L., et al. (2016) extracted the anomalies of the fluid data by calculating $\lambda^2$.

[Figure]

图 6 汶川 8.0 级地震前流体高频信息异常曲线

Fig. 6 Curves of high-frequency fluid anomaly information before Wenchuan $M_S 8.0$ earthquake

Yan R., et al. (2011) extracted the anomalies of the radon concentrations by calculating AR(1) coefficient (they defined).

[Figure]

图 10 汶川 8.0 级地震前中国大陆水氡存在临界慢化现象的 AR(1)系数曲线
(括号中的数字表示距离汶川 8.0 级主震震中的距离)

Fig. 10 AR(1) coefficient curves of water radon for the existence of critical slowing down phenomenon before Wenchuan 8.0 earthquake in China mainland. (Numbers in parentheses denote the distance from epicenter of Wenchuan 8.0 main earthquake)

We are very grateful to the reviewers and editors for their contributions to improving the quality of our manuscript. We have revised the manuscript according to your comments. The response to each revision is listed as follows:

**Response to report 1:**

The authors have done a good job in responding the comments and modifying the manuscript. By adding Sec.5 where they compare their findings for further sensors and other periods (non disturbed by large shocks), they provide further evidence to strengthen the unusual character of their observations. They also ruled out a possible influence of environmental factors on the strain signals during the 2008 Wenchuan earthquake. I agree with their conclusion which stated that although the negentropy method and its results are interesting (and intriguing), linking the observed changes to the nucleation process still requires further data and analyses. I therefore recommand publication in NPG journal.

One point can still be improved concerning the description of the data (Sec. 2) and particularly how the authors derive the 'residual high frequency signals' (in Fig. 3). The authors explained clearly their approach in the response letter to reviewer 2 (as steps 1 & 2, pp. 9-10), also it would be interesting if they can also detailed the protocol (basically steps 1 & 2) in the manuscript.

**Changes:**
We have detailed the how we derive the residual high frequency signals through Step 1 and Step 2 in Line 69-87.

**Minor comments:**
- Abstract (L. 2) : you can say that you analyze negentropy for 3 strainmeters (not only GUZA) ?
We have modified.
- Abstract (L. 7) : 'Combined with the confusion discussion …' : what you refer here is not clear, maybe just provide major results of this section in the Abstract ?
We have modified the Abstract.
- Introduction (L. 10) : 'at least some large' → at least **for** some large.
We have modified.
- Introduction (L. 13) : Hsu et al. (2015) paper does not analyze earthquake strain signals (only meteorological perturbations), there may be a confusion here?
Yes, Hsu et al. didn't focus on earthquake strain signals. But they extended an investigation with a quantitative analysis of the strain responses to precipitation as well as barometric pressure and the earth tides in order to isolate tectonic source effects. And they analyzed whether corrected borehole strain changes are related to environmental disturbances or tectonic-original motions. Therefore, we think the borehole strain observations were of research significance.
However, combined with the context, this reference may be a little confusion. We have removed this reference.
- Introduction (L. 50) : the authors may consider changing the name of the 'Confusion Discussion' (Sec. 5), that sounds awkward when reading the main text.
We have modified as "Comparison Discussion".

- Fig. 1 (legend) : please consider adding details (e.g., the blue rectangles show the strainmeter stations, the epicenter is shown by yellow star + add the timing of the mainshock, 12 May 2008).

We have modified.

- L. 184-185 : 'Thus, we consider the Wenchuan earthquake day may be a critical time during the whole Wenchuan earthquake process' → that sentence is weird and has no clear purpose, you may consider remove it.

We have removed it.

L. 196-199 : these sentences are confusing, please consider rewrite them.

We have rewritten these sentences.

L. 218-219 : 'However, since the curve is approximately, … the value of inflection point exists a range.' → which range ? Something is missing in this sentence.

We have modified.

L. 219 : may **be** able.

We have modified.

L. 219 and 224 : 'receive' → rather 'detect' or 'record'.

We have modified.

L. 243 :'Through the confusion discussion' → modify by 'In Sec. 5' for exemple.

We have modified as "Comparison Discussion".

**Response to report 2:**

**General Comments:**

All the Authors' corrections and replies are in the right direction.

- Probably there is a misunderstanding concerning my first comment. I did not propose a whole new paragraph about the "Negentropy" notion but about the "borehole strain" method ! However it has been a good idea to add a new paragraph about the "Negentropy" notion !

- The NPG paper by Karamanos et al. (NPG, 2005) mainly uses the "block-entropy analysis by lumping" of symbolic sequences, formally introduced in the literature in: "Entropy analysis of substitutive sequences revisited." K. Karamanos J. Phys. A: Math. Gen. 34, 9231-9241 (2001).

- The Phys. Rev. E paper by Karamanos et al. (Phys. Rev. E, 2006) mainly uses the "T-complexity" notion to preseismic precursors for the first time in the international literature of Geophysics.

- The notion of the approximate Entropy (ApEn) is mainly used in: "A unified approach of catastrophic events." S. Nikolopoulos, P. Kapiris, K. Karamanos and K. Eftaxias Nat. Haz. Earth Syst. Sciences 4, 615-631 (2004).

Of course, I leave to the discretion of the Editor and the Authors to add or not the above mentioned References.

**Response:**

We are very grateful to the reviewer for providing these references on the first use of some notions, which gives us a deeper understanding of entropy.

However, these documents and those we have quoted are a little repetitive for our manuscript.

Therefore, we have only supplemented one of them which is most helpful to our manuscript.

**References:**

[revised manuscript text omitted]